# Online Deflection Compensation of a Flexible Hydraulic Loader Crane Using Neural Networks and Pressure Feedback

**Konrad Johan Jensen** * [ID], **Morten Kjeld Ebbesen** [ID] and **Michael Rygaard Hansen** [ID]

Department of Engineering Sciences, University of Agder, 4879 Grimstad, Norway;
morten.k.ebbesen@uia.no (M.K.E.); michael.r.hansen@uia.no (M.R.H.)
* Correspondence: konrad.j.jensen@uia.no

**Abstract:** The deflection compensation of a hydraulically actuated loader crane is presented. Measurement data from the laboratory are used to design a neural network deflection estimator. Kinematic expressions are derived and used with the deflection estimator in a feedforward topology to compensate for the static deflection. A dynamic deflection compensator is implemented, using pressure feedback and an adaptive bandpass filter. Simulations are conducted to verify the performance of the control system. Experimental results showcase the effectiveness of both the static and dynamic deflection compensator while running closed-loop motion control, with a 90% decrease in static deflection.

**Keywords:** deflection compensation; kinematics; loader crane; hydraulics; neural network

## 1. Introduction

Flexible manipulators have received extensive research attention in recent years. The use of lightweight though flexible manipulators yields many advantages over rigid structures, including lower mass and inertia, lower energy consumption, higher payload-to-weight ratio, and smaller actuators. However, there are challenges associated with the structural flexibility of these manipulators that must be taken into account. The deflection, oscillations, and potential nonlinearities may lead to issues with steady-state performance, stability, and controllability.

Different approaches of modeling flexible manipulators have previously been considered, such as lumped parameter [1–3], assumed modes [4–6], Lagrangian formulation [7,8] and neural networks [9].

The control of flexible manipulators is typically divided into two groups, model-based control and model-free control. The primary goal of both control techniques is to dampen oscillations and reduce the consequences of static deflection in the flexible manipulator. Model-based control may use the modeling techniques shown earlier, and can be implemented in a feedforward topology. This includes control with linear models [10], nonlinear inverse dynamics [11], and input shaping [12]. In model-free control, the system does not rely on a mathematical model of the system, but rather sensor measurements from the system. Model-free control includes robust control and sliding mode control [13–15].

Another technique that has received research interest for the control of flexible manipulators is neural network control. This can include both feedforward and feedback controllers [16]. Neural networks are often combined with sliding mode control for robust control and the stabilization of nonlinear systems [17–19]. Kinematic control of redundant manipulators was investigated in [20,21].

Large manipulators, such as hydraulic cranes, may experience large static deflections under heavy load. This is especially an issue with weight-optimized structures, such as loader cranes. As a consequence, the calculated crane tip position based on rigid body kinematics may yield significant errors and may be a safety concern that, potentially, can lead to collisions with the surroundings if not compensated for. This is especially

true when using closed-loop motion control and when forward kinematics are used to estimate the crane tip position. For manually operated cranes, the operator may visually identify and compensate for the deflections, effectively closing the loop. This is often called operator in the loop. To reach the same level of automation as industrial robots, deflection compensation may play a critical role in increasing the precision and safety when using closed-loop motion control for large cranes.

In this paper, a new method for closed-loop control of a hydraulic manipulator is presented. The novelty lies in the combination of compensation for both static and dynamic deflection while running path control. This combination of path control and static and dynamic compensation is an answer to the previously mentioned problems for highly flexible manipulators, and is therefore developed and implemented on a commercial hydraulically actuated loader crane.

## 2. Considered System

In this paper, a HMF 2020K4 loader crane made by HMF Group A/S (Højbjerg, Denmark) is used as a basis for simulations and experiments; see Figure 1. The crane has three degrees of freedom of interest: the rotation of the main boom, the rotation of the knuckle boom and the extension of the telescopic booms. They are controlled by means of the main cylinder, the knuckle cylinder and, working as a single sequential cylinder, the telescopic cylinders. Each cylinder is driven by a pressure-compensated directional control valve, which ensures load-independent flow control. Counterbalance valves are used for load holding, assisting in load lowering, and protection against pressure surges. An illustration of the hydraulic system for the knuckle cylinder is shown in Figure 2.

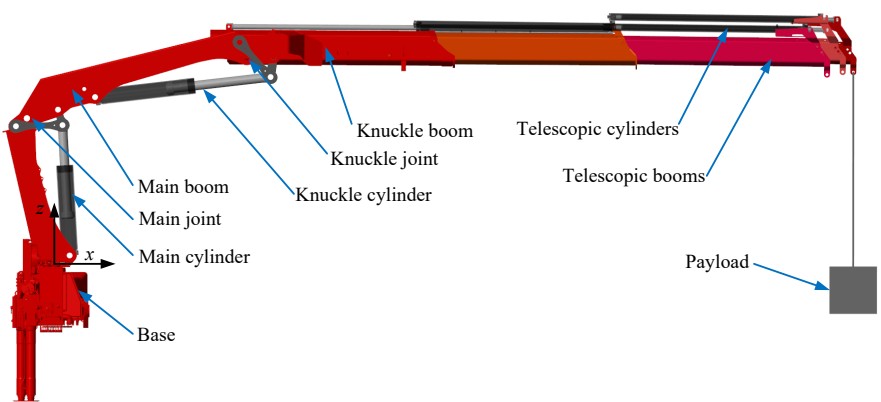

**Figure 1.** Illustration of the HMF 2020K4 loader crane.

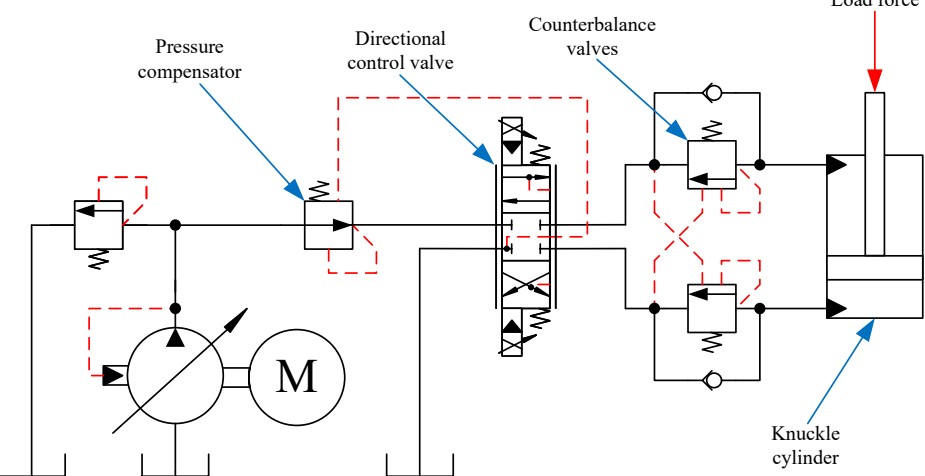

**Figure 2.** Hydraulic system of the knuckle cylinder.

*Control Strategy*

The novel approach is shown in Figure 3. The crane is running path control in the actuator space with position feedback and velocity feedforward developed in [22]. The control strategy for the deflection compensation is split into static compensation and dynamic compensation. The static deflection compensator uses feedforward and adjusts the position reference based on an estimated deflection for a given actuator position. The dynamic deflection compensator uses feedback of the load pressure $p_L$ to measure and suppress the oscillations of the crane.

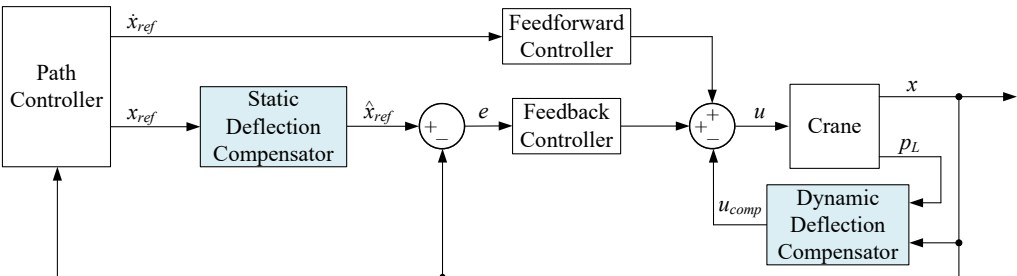

**Figure 3.** Control strategy with the novel deflection compensators highlighted in light blue.

## 3. Static Deflection Compensation

The static deflection compensator is a model-based feedforward controller and is based on a deflection estimator and kinematic functions. The estimated deflection of the crane is in Cartesian space, while the motion controller operates in actuator space. The relevant kinematic relations are derived in this section and are then used to generate a modified cylinder position reference.

### 3.1. Measuring Deflection in Laboratory

Experiments are conducted in the laboratory using a laser tracker, namely a Leica Absolute Tracker AT960. The laser tracker measures the position of a reflector mounted on the crane tip. A 581 kg payload is connected to the winch on the crane, and by measuring the crane tip position with and without the payload, the deflection of the crane tip is effectively measured. The setup in the laboratory is shown in Figure 4. Note that 581 kg was the heaviest payload that the crane could consistently lift at this boom length and pressure level available in the laboratory.

Multiple measurements are conducted with different cylinder positions. The resulting crane tip position in the *xz*-plane with and without load is shown in Figure 5, with the black lines illustrating the crane position for one of the samples. Deflection is calculated as the difference between the load and no-load tip position.

### 3.2. Forward Kinematics

Forward kinematics are used to go from joint space to Cartesian space. The forward kinematics are calculated based on Denavit–Hartenberg parameters. Figure 6 shows the joint angles, telescopic length, lifting radius, and tip position. Figure 7 shows the geometry which is used with the Denavit–Hartenberg parameters, where both booms are horizontal. The dimensions between consecutive joints are shown in Table 1. The Denavit–Hartenberg parameters are shown in Table 2, where **R** and **T** are rotational and translational matrices, respectively. The angles $\theta_m$ and $\theta_k$ denote the rotation about the main joint and knuckle joint, respectively. The forward kinematics are similar to what was developed earlier in [23], with the addition of the telescopic actuator length $x_t$ used in this paper.

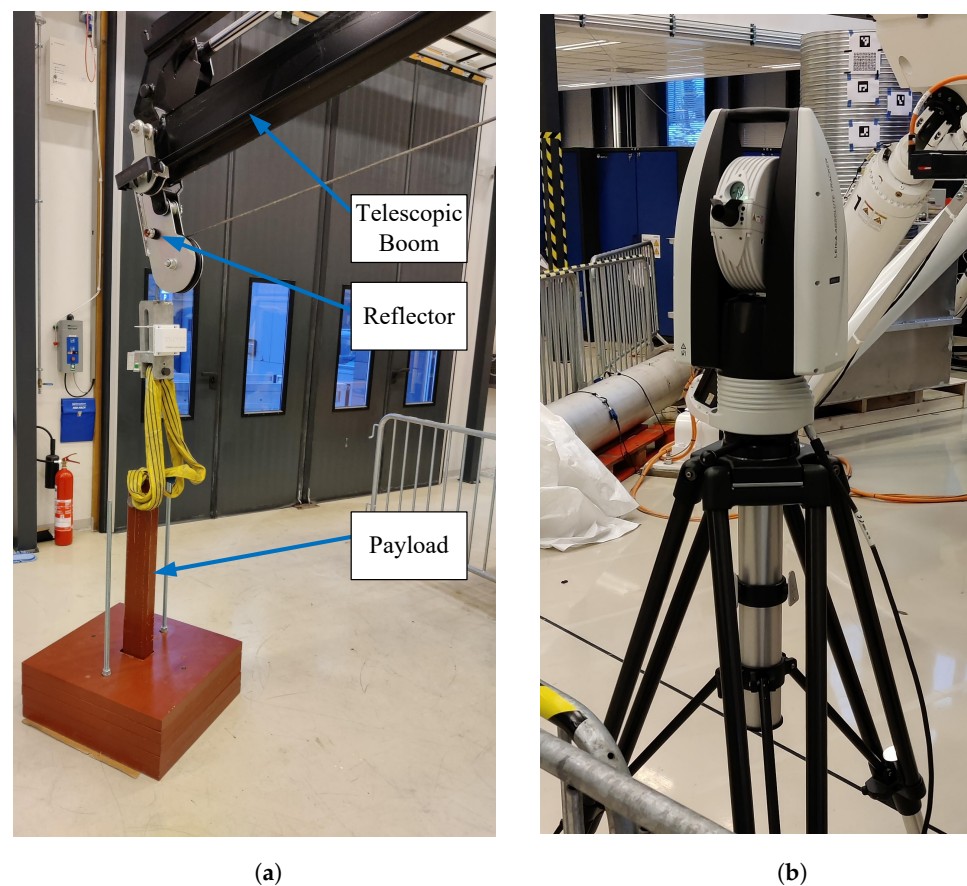

(a)                                                       (b)

**Figure 4.** Experimental setup in the lab. The laser tracker measures the crane tip position using the attached reflector. (**a**) Crane tip showing the telescopic boom, reflector, and payload, (**b**) Leica Absolute Tracker AT960.

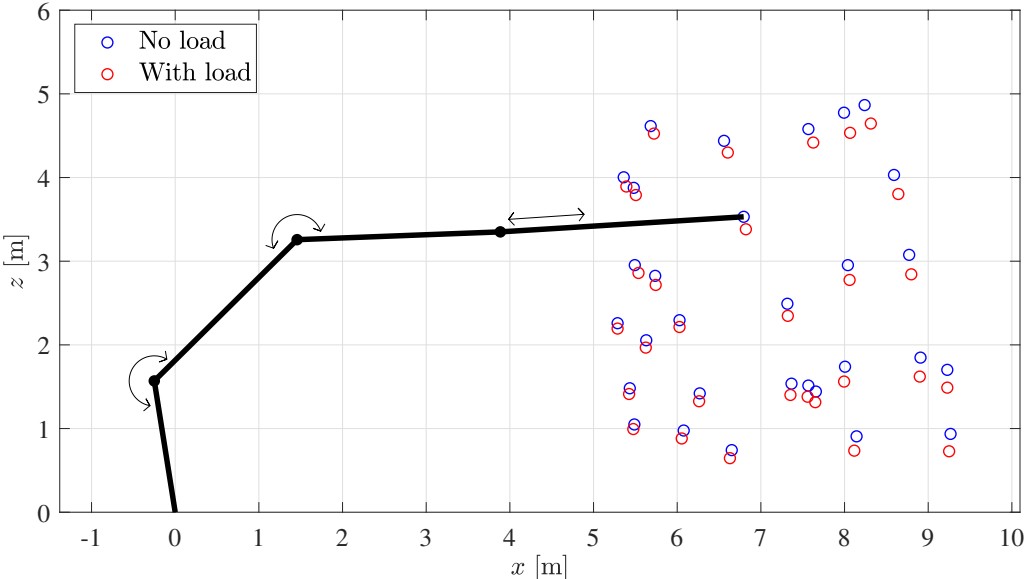

**Figure 5.** Crane tip position in *xz*-plane with and without load from laboratory measurements. Crane position illustrated in black with its three degrees of freedom.

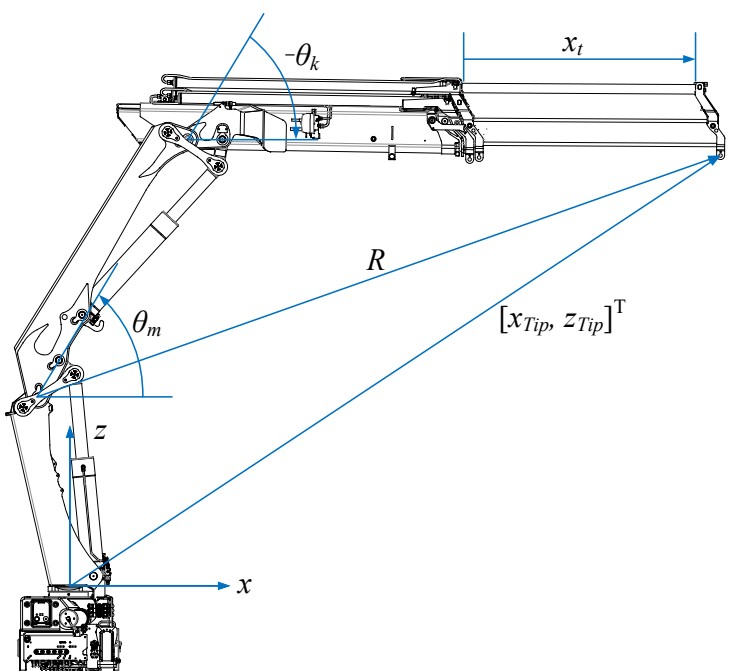

**Figure 6.** Crane geometry showing joint angles, lifting radius $R$, telescopic length $x_t$, and crane tip position.

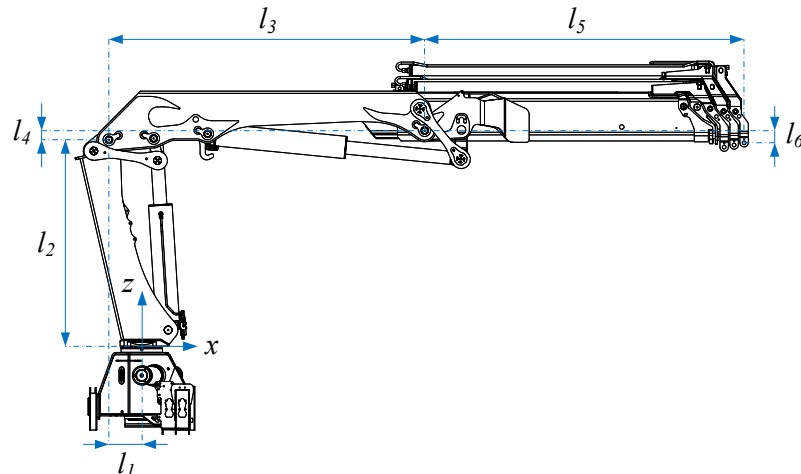

**Figure 7.** Crane geometry used with Denavit–Hartenberg parameters.

**Table 1.** Dimensions shown in Figure 7.

| Name | Length [m] |
|------|------------|
| $l_1$ | 0.250 |
| $l_2$ | 1.569 |
| $l_3$ | 2.400 |
| $l_4$ | 0.070 |
| $l_5$ | 2.429 |
| $l_6$ | 0.093 |

**Table 2.** Denavit–Hartenberg parameters.

| $\mathbf{R}_z$ | $\mathbf{T}_z$ | $\mathbf{T}_x$ | $\mathbf{R}_x$ |
|:---:|:---:|:---:|:---:|
| 0 | $l_2$ | $-l_1$ | $90°$ |
| $\theta_m$ | 0 | 0 | $-90°$ |
| 0 | $l_4$ | $l_3$ | $90°$ |
| $\theta_k$ | 0 | 0 | $-90°$ |
| 0 | $-l_6$ | $l_5$ | 0 |
| 0 | 0 | $x_t$ | 0 |

The transformation matrix $\mathbf{A}_{DH}$ is given as

$$\mathbf{A}_{DH} = \mathbf{T}_z(l_2) \cdot \mathbf{T}_x(-l_1) \cdot \mathbf{R}_x(90°) \cdot \mathbf{R}_z(\theta_m) \cdot \mathbf{R}_x(-90°) \cdot \mathbf{T}_z(l_4) \cdot \mathbf{T}_x(l_3) \cdot \mathbf{R}_x(90°)$$
$$\cdot \mathbf{R}_z(\theta_k) \cdot \mathbf{R}_x(-90°) \cdot \mathbf{T}_z(-l_6) \cdot \mathbf{T}_x(l_5) \cdot \mathbf{T}_x(x_t) \tag{1}$$

$$= \begin{bmatrix} c_{\theta_m+\theta_k} & 0 & -s_{\theta_m+\theta_k} & x_{Tip} \\ 0 & 1 & 0 & 0 \\ s_{\theta_m+\theta_k} & 0 & c_{\theta_m+\theta_k} & z_{Tip} \\ 0 & 0 & 0 & 1 \end{bmatrix} \tag{2}$$

The crane tip positions $x_{Tip}$ and $z_{Tip}$ are given in Equations (3) and (4), using the notation $\cos(\theta) = c_\theta$ and $\sin(\theta) = s_\theta$.

$$x_{Tip} = -l_1 + l_3 \cdot c_{\theta_m} - l_4 \cdot s_{\theta_m} + l_5 \cdot c_{\theta_m+\theta_k} + l_6 \cdot s_{\theta_m+\theta_k} + x_t \cdot c_{\theta_m+\theta_k} \tag{3}$$
$$z_{Tip} = l_2 + l_3 \cdot s_{\theta_m} + l_4 \cdot c_{\theta_m} + l_5 \cdot s_{\theta_m+\theta_k} - l_6 \cdot c_{\theta_m+\theta_k} + x_t \cdot s_{\theta_m+\theta_k} \tag{4}$$

*3.3. Inverse Kinematics*

Inverse kinematics are used to go from Cartesian space to joint space. Solving the inverse kinematics of the crane is similar to a typical two-link manipulator, with the exception that the length of each link is split into an $x$-component and $z$-component. In addition, the three actuators give the crane kinematic redundancy in the case of motion in the $xz$-plane. This is solved by keeping the telescopic actuator length $x_t$ fixed and solving for the main joint angle $\theta_m$ and knuckle joint angle $\theta_k$. This is done because the main cylinder and knuckle cylinder are easier to control and have less friction than the telescopic cylinder.

The calculations are based on the lifting radius $R$, which is the distance from the main joint to the crane tip. The squared lifting radius $R^2$ is given by

$$R^2 = \left(x_{Tip} + l_1\right)^2 + \left(z_{Tip} - l_2\right)^2 \tag{5}$$

Some intermediate equations are used to solve for the knuckle boom angle $\theta_k$. Inserting Equations (3) and (4) into (5) yields

$$\begin{aligned} R^2 &= (l_3 \cdot c_{\theta_m} - l_4 \cdot s_{\theta_m} + l_5 \cdot c_{\theta_m+\theta_k} + l_6 \cdot s_{\theta_m+\theta_k} + x_t \cdot c_{\theta_m+\theta_k})^2 \\ &\quad + (l_3 \cdot s_{\theta_m} + l_4 \cdot c_{\theta_m} + l_5 \cdot s_{\theta_m+\theta_k} - l_6 \cdot c_{\theta_m+\theta_k} + x_t \cdot s_{\theta_m+\theta_k})^2 \\ &= 2 \cdot (l_3 \cdot l_5 - l_4 \cdot l_6 + l_3 \cdot x_t) \cdot c_{\theta_k} + 2 \cdot (l_3 \cdot l_6 + l_4 \cdot l_5 + l_4 \cdot x_t) \cdot s_{\theta_k} \\ &\quad + l_3^2 + l_4^2 + l_5^2 + l_6^2 + 2 \cdot l_5 \cdot x_t + x_t^2 \end{aligned} \tag{6}$$

The equations in a more compact form are given below:

$$R^2 = A \cdot c_{\theta_k} + B \cdot s_{\theta_k} + C \tag{7}$$
$$A = 2 \cdot (l_3 \cdot l_5 - l_4 \cdot l_6 + l_3 \cdot x_t) \tag{8}$$
$$B = 2 \cdot (l_3 \cdot l_6 + l_4 \cdot l_5 + l_4 \cdot x_t) \tag{9}$$
$$C = l_3^2 + l_4^2 + l_5^2 + l_6^2 + 2 \cdot l_5 \cdot x_t + x_t^2 \tag{10}$$

Solving Equation (7) yields two solutions, and by taking the minimum angle, the crane will be in the desired elbow-up configuration. The calculation of $\theta_k$ is shown below:

$$\theta_k^* = 2 \cdot \tan^{-1}\left( \frac{B \pm \sqrt{A^2 + B^2 - C^2 + 2 \cdot C \cdot R^2 - R^4}}{A - C + R^2} \right) \tag{11}$$

$$\theta_k = \min(\theta_k^*) \tag{12}$$

To find $\theta_m$, Equations (3) and (4) are expanded, and the terms containing $\theta_m$ are factorized out.

$$x_{Tip} = (l_5 \cdot c_{\theta_k} + l_6 \cdot s_{\theta_k} + x_t \cdot c_{\theta_k} + l_3) \cdot c_{\theta_m} - (l_5 \cdot s_{\theta_k} - l_6 \cdot c_{\theta_k} + x_t \cdot s_{\theta_k} + l_4) \cdot s_{\theta_m} - l_1 \tag{13}$$

$$z_{Tip} = (l_5 \cdot s_{\theta_k} - l_6 \cdot c_{\theta_k} + x_t \cdot s_{\theta_k} + l_4) \cdot c_{\theta_m} + (l_5 \cdot c_{\theta_k} + l_6 \cdot s_{\theta_k} + x_t \cdot c_{\theta_k} + l_3) \cdot s_{\theta_m} + l_2 \tag{14}$$

Rearranging gives a more compact form, yielding two equations with two unknowns, namely $\cos(\theta_m)$ and $\sin(\theta_m)$.

$$x_{Tip} = E \cdot c_{\theta_m} - D \cdot s_{\theta_m} - l_1 \tag{15}$$

$$z_{Tip} = D \cdot c_{\theta_m} + E \cdot s_{\theta_m} + l_2 \tag{16}$$

$$D = l_5 \cdot s_{\theta_k} - l_6 \cdot c_{\theta_k} + x_t \cdot s_{\theta_k} + l_4 \tag{17}$$

$$E = l_5 \cdot c_{\theta_k} + l_6 \cdot s_{\theta_k} + x_t \cdot c_{\theta_k} + l_3 \tag{18}$$

These two equations are then solved to find $\theta_m$.

$$c_{\theta_m} = \frac{D \cdot z_{Tip} - D \cdot l_2 + E \cdot x_{Tip} + E \cdot l_1}{D^2 + E^2} \tag{19}$$

$$s_{\theta_m} = \frac{E \cdot z_{Tip} - E \cdot l_2 - D \cdot x_{Tip} - D \cdot l_1}{D^2 + E^2} \tag{20}$$

$$\theta_m = \tan^{-1}\left( \frac{s_{\theta_m}}{c_{\theta_m}} \right) \tag{21}$$

### 3.4. Actuator Kinematics

Actuator kinematics are used to go from actuator space to joint space. They were developed earlier in [23]. An illustration of the main joint linkage is shown in Figure 8. The associated lengths are given in Table 3.

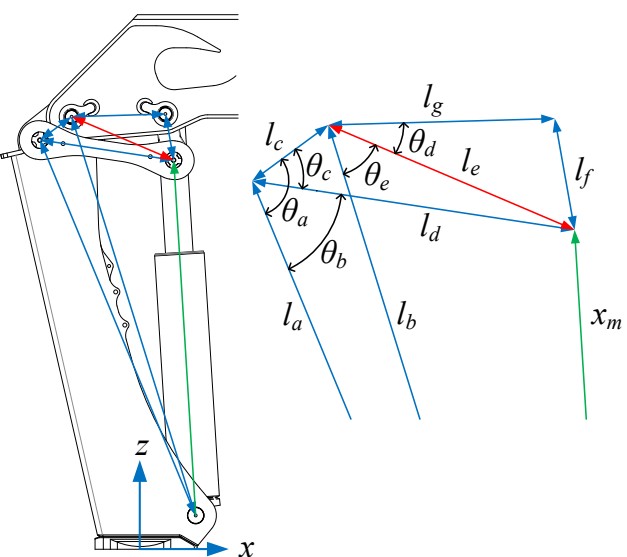

**Figure 8.** Illustration of the main joint actuator kinematics.

**Table 3.** Lengths of the parts in the main linkage.

| Name | Length [m] |
|---|---|
| $l_a$ | 1.473 |
| $l_b$ | 1.514 |
| $l_c$ | 0.143 |
| $l_d$ | 0.490 |
| $l_f$ | 0.170 |
| $l_g$ | 0.340 |

For reference, the calculation of the main joint angle $\theta_m = \theta_m(x_m)$ is given below:

$$\theta_a = \cos^{-1}\left(\frac{l_a^2 + l_c^2 - l_b^2}{2 \cdot l_a \cdot l_c}\right) \tag{22}$$

$$\theta_b = \cos^{-1}\left(\frac{l_a^2 + l_d^2 - x_m^2}{2 \cdot l_a \cdot l_d}\right) \tag{23}$$

$$\theta_c = \theta_a - \theta_b \tag{24}$$

$$l_e = \sqrt{l_c^2 + l_d^2 - 2 \cdot l_c \cdot l_d \cdot c_{\theta_c}} \tag{25}$$

$$\theta_d = \cos^{-1}\left(\frac{l_e^2 + l_g^2 - l_f^2}{2 \cdot l_e \cdot l_g}\right) \tag{26}$$

$$\theta_e = \cos^{-1}\left(\frac{l_b^2 + l_e^2 - x_m^2}{2 \cdot l_b \cdot l_e}\right) \tag{27}$$

$$\theta_m = \theta_d + \theta_e - \tilde{\theta}_m \tag{28}$$

*3.5. Inverse Actuator Kinematics*

Inverse actuator kinematics are used to go from joint space to actuator space. As taking the inverse of Equation (28) is difficult due to it being such a complex expression, curve fitting is used instead. The 9th order polynomials are used, shown below:

$$x = p_9 \cdot \theta^9 + p_8 \cdot \theta^8 + p_7 \cdot \theta^7 + p_6 \cdot \theta^6 + p_5 \cdot \theta^5 + p_4 \cdot \theta^4 + p_3 \cdot \theta^3 + p_2 \cdot \theta^2 + p_1 \cdot \theta + p_0 \tag{29}$$

While the mapping $\theta_m = \theta_m(x_m)$ given in Equation (28) describes the actuator kinematics for the main cylinder, the inverse kinematics is the mapping $x_m = x_m(\theta_m)$. Iteratively calculating and plotting $\theta_m = \theta_m(x_m)$ and then switching the axis gives a solution on which the curve is fitted. The coefficients for the main cylinder and knuckle cylinder are given in Table 4, and plots of the curve fits are shown in Figure 9, showing that the curve fit yields a close match to the numerical inverse.

**Table 4.** Curve-fitting coefficients for inverse actuator kinematics.

| Coefficient | Main | Knuckle |
|---|---|---|
| $p_9$ | $-8.324 \times 10^{-5}$ | $-2.044 \times 10^{-5}$ |
| $p_8$ | $4.068 \times 10^{-4}$ | $-2.996 \times 10^{-4}$ |
| $p_7$ | $-4.087 \times 10^{-4}$ | $-1.571 \times 10^{-3}$ |
| $p_6$ | $-1.797 \times 10^{-3}$ | $-4.609 \times 10^{-3}$ |
| $p_5$ | $2.914 \times 10^{-3}$ | $-1.045 \times 10^{-2}$ |
| $p_4$ | $1.293 \times 10^{-2}$ | $-1.135 \times 10^{-2}$ |
| $p_3$ | $-4.794 \times 10^{-2}$ | $3.451 \times 10^{-3}$ |
| $p_2$ | $2.438 \times 10^{-2}$ | $1.153 \times 10^{-2}$ |
| $p_1$ | $3.471 \times 10^{-1}$ | $3.042 \times 10^{-1}$ |
| $p_0$ | $1.291$ | $1.923$ |

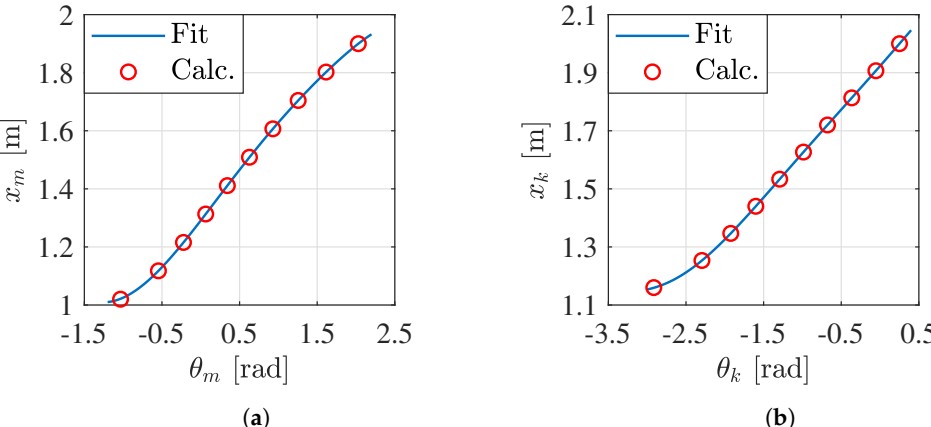

(**a**)　　　　　　　　　　　　　　(**b**)

**Figure 9.** Curve fit for inverse actuator kinematics. (**a**) Curve fit for main cylinder, (**b**) Curve fit for knuckle cylinder.

### 3.6. Neural Network Deflection Estimator

A neural network is used to estimate the deflection of the crane tip. Measurements from the laboratory make up the training data for the network. The network predicts the deflection in the $x$- and $z$-directions based on the cylinder positions $x_m$, $x_k$, and $x_t$. The selected topology is a classical multilayer perceptron with a single hidden layer. Each node uses the tanh activation function. Input scaling is employed to normalize the data to the range $[-1, 1]$, in order to stay in the center region of the tanh activation function. Likewise, the output scaling is used to scale the outputs from $[-1, 1]$ to a desired range, set by the measured deflection in the output training data. An illustration of a single node with weights, bias, and activation function is shown in Figure 10. An overview of the neural network with input scaling, output scaling, and ten hidden neurons is shown in Figure 11.

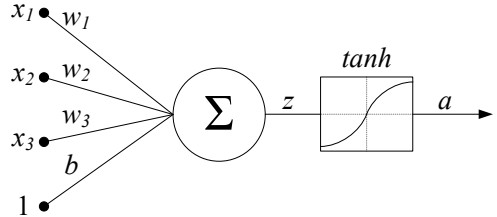

**Figure 10.** Illustration of a single node.

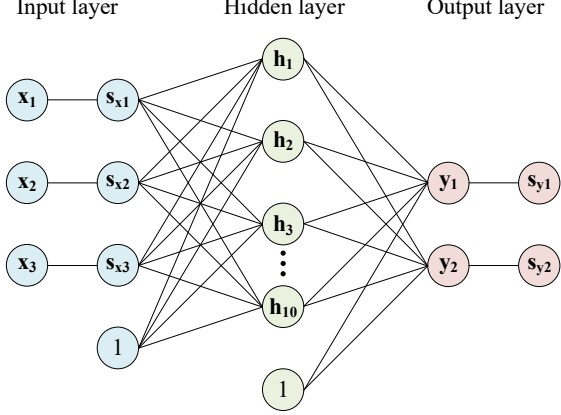

**Figure 11.** Overview of the neural network.

### 3.6.1. Forward Propagation

Forward propagation refers to computing the outputs of the network with a given input. The computations are done using vectors and matrices to calculate the output of each layer in a single step. The formula for the input scaling is shown in Equations (30)–(34). The input is scaled to lie between $-1$ and $1$ based on the maximum and minimum value of the input training data $\mathbf{x}_{training}$.

$$\mathbf{s_x} = (\mathbf{x} - \mathbf{x}_{min}) \cdot \frac{\mathbf{s}_{x,max} - \mathbf{s}_{x,min}}{\mathbf{x}_{max} - \mathbf{x}_{min}} + \mathbf{s}_{x,min} \tag{30}$$

$$\mathbf{x}_{min} = \min(\mathbf{x}_{training}) \tag{31}$$

$$\mathbf{x}_{max} = \max(\mathbf{x}_{training}) \tag{32}$$

$$\mathbf{s}_{x,min} = -1 \tag{33}$$

$$\mathbf{s}_{x,max} = 1 \tag{34}$$

Forward propagation for the hidden layer and output layer using tanh as the activation function is shown in Equations (35)–(38).

$$\mathbf{z_h} = \mathbf{W_h} \cdot \mathbf{s_x} + \mathbf{b_h} \tag{35}$$

$$\mathbf{h} = \tanh(\mathbf{z_h}) \tag{36}$$

$$\mathbf{z_y} = \mathbf{W_y} \cdot \mathbf{h} + \mathbf{b_y} \tag{37}$$

$$\mathbf{y} = \tanh(\mathbf{z_y}) \tag{38}$$

where

$\mathbf{W_h}$ = weight matrix of the hidden layer;
$\mathbf{b_h}$ = bias vector of the hidden layer;
$\mathbf{W_y}$ = weight matrix of the output layer;
$\mathbf{b_y}$ = bias vector of the output layer.

The output scaling is similar to the input scaling, and is shown in Equations (39)–(43).

$$\mathbf{s_y} = (\mathbf{y} - \mathbf{y}_{min}) \cdot \frac{\mathbf{s}_{y,max} - \mathbf{s}_{y,min}}{\mathbf{y}_{max} - \mathbf{y}_{min}} + \mathbf{s}_{y,min} \tag{39}$$

$$\mathbf{y}_{min} = -1 \tag{40}$$

$$\mathbf{y}_{max} = 1 \tag{41}$$

$$\mathbf{s}_{y,min} = \min(\mathbf{y}_{training}) \tag{42}$$

$$\mathbf{s}_{y,max} = \max(\mathbf{y}_{training}) \tag{43}$$

### 3.6.2. Backpropagation

Backpropagation refers to the process of calculating the gradient of the cost function with respect to the weights. This is typically done using the chain rule one layer at the time. The gradient descent is then used to update the weights. The training data are now a matrix, where each column is a single measurement. The cost function is made using the squared Frobenius norm of the scaled output minus the output training data. The cost function is defined as

$$C = \frac{1}{2} \cdot \left\| \mathbf{s_y} - \mathbf{y}_{training} \right\|_F^2 \tag{44}$$

To train the network, the partial derivatives of the cost function must first be calculated. Note that the derivative of the activation function is $\frac{d}{dx} \tanh(x) = 1 - \tanh^2(x)$. The backpropagation for the output layer and the weight $\mathbf{W_h}$ is shown in Equations (45)–(49). Di-

viding by the number of training examples, $N$ is used to average the calculations across the training set.

$$\frac{\partial C}{\partial \mathbf{y}} = \mathbf{s_y} - \mathbf{y}_{training} \tag{45}$$

$$\frac{\partial \mathbf{y}}{\partial \mathbf{z_y}} = 1 - \tanh^2(\mathbf{z_y}) \tag{46}$$

$$\frac{\partial C}{\partial \mathbf{z_y}} = \frac{\partial C}{\partial \mathbf{y}} \cdot \frac{\partial \mathbf{y}}{\partial \mathbf{z_y}} \tag{47}$$

$$\frac{\partial \mathbf{z_y}}{\partial \mathbf{W_y}} = \mathbf{h}^T \tag{48}$$

$$\frac{\partial C}{\partial \mathbf{W_y}} = \frac{1}{N} \cdot \frac{\partial C}{\partial \mathbf{z_y}} \cdot \frac{\partial \mathbf{z_y}}{\partial \mathbf{W_y}} \tag{49}$$

The partial derivatives are then used to update the weights. L2 regularization is used to avoid overfitting. This limits the value of the weights in the network to achieve better generalization. Updating the weights with L2 regularization only requires one additional parameter $\lambda$, in addition to the learning rate $\eta$. The adjusted cost function and the updates to the weight $\mathbf{W_h}$ are shown in Equations (50)–(51).

$$C^* = C + \frac{\lambda}{2 \cdot N} \cdot \|\mathbf{W_h}\|_F^2 \tag{50}$$

$$\mathbf{W_h} \leftarrow \mathbf{W_h} - \eta \cdot \left( \frac{\partial C}{\partial \mathbf{W_h}} + \lambda \cdot \mathbf{W_h} \right) \tag{51}$$

### 3.6.3. Training Results

The measured deflection from the laboratory is used to train the network. The network is trained using $\eta = 1$ and $\lambda = 10^{-4}$ and converged after $3 \times 10^5$ iterations. Surface plots of the estimated deflection for some cylinder lengths are shown in Figures 12 and 13. It can be seen that the telescope length $x_t$ does not significantly impact the deflection in $x$-direction, but it has a major contribution to the deflection in the $z$-direction.

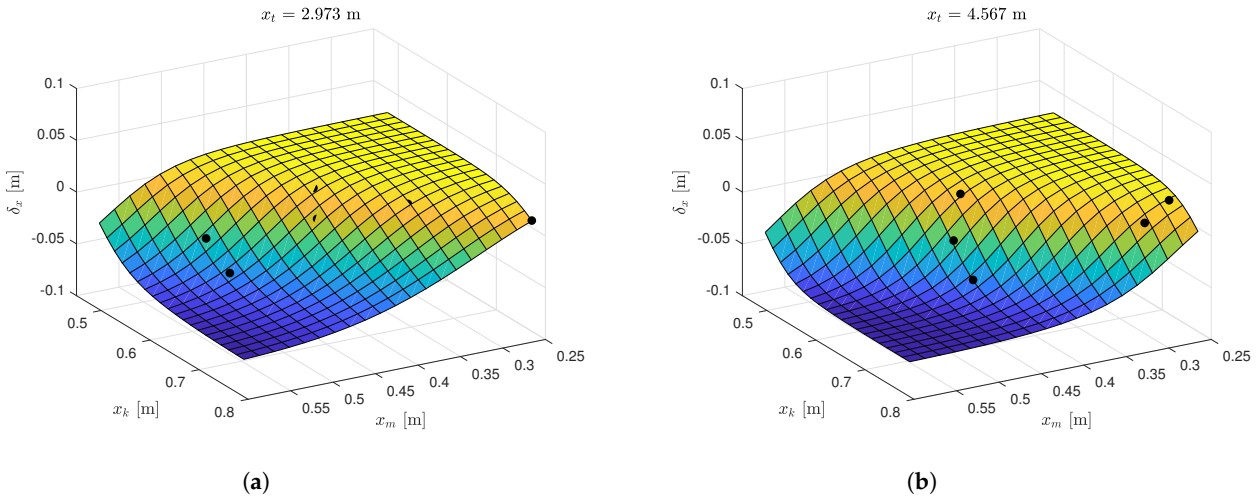

(**a**)　　　　　　　　　　　　　　　　　　(**b**)

**Figure 12.** Predicted deflection in $x$-direction $\delta_x$. Black dots show measured data from the laboratory. (**a**) Predicted $\delta_x$ with $x_t = 2.973$ m. (**b**) Predicted $\delta_x$ with $x_t = 4.567$ m.

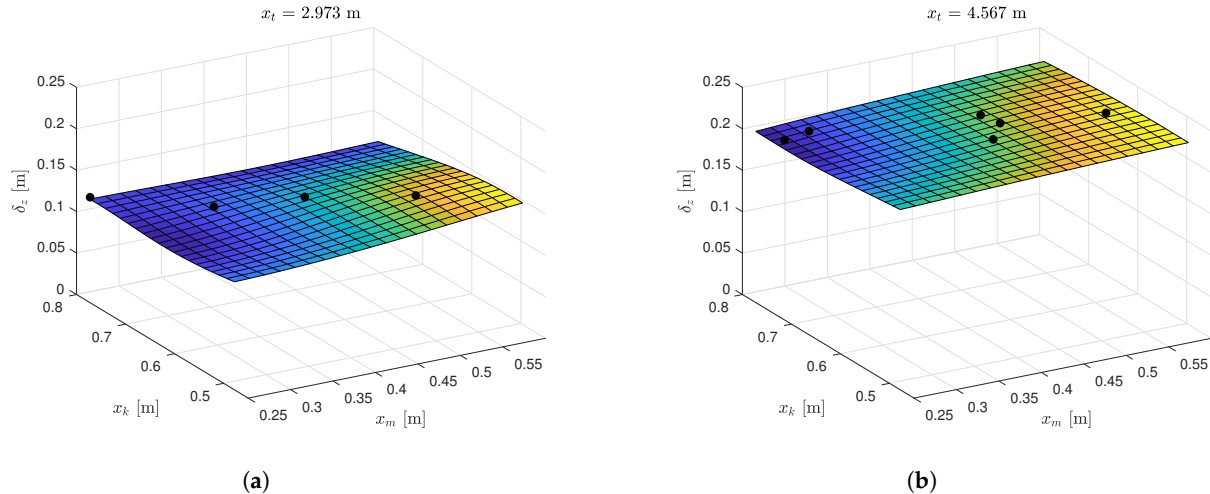

(**a**)                                                            (**b**)

**Figure 13.** Predicted deflection in *z*-direction $\delta_z$. Black dots show measured data from the laboratory. (**a**) Predicted $\delta_z$ with $x_t = 2.973$ m. (**b**) Predicted $\delta_z$ with $x_t = 4.567$ m.

### 3.7. Control System

The block diagram for the static deflection compensator is shown in Figure 14. The system uses actuator kinematics (Act. Kin.) and forward kinematics (For. Kin.) to transform the cylinder position references into Cartesian space. The output of the deflection estimator (Def. Est.) is added to the Cartesian position reference. Inverse kinematics (Inv. Kin.) and inverse actuator kinematics (Inv. Act.) are then used to generate the modified cylinder position references $\hat{x}_{ref}$. It should be noted that while the telescopic position reference is used in the calculations, it is not modified since only the main cylinder and knuckle cylinder compensate for the deflection.

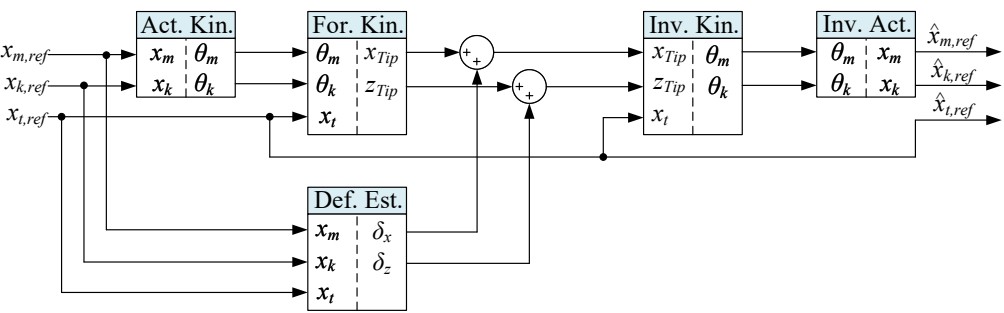

**Figure 14.** Block diagram of the static deflection compensator.

## 4. Dynamic Deflection Compensation

The dynamic deflection compensator is based on the feedback of the load pressure on the main cylinder. Pressure feedback has previously shown its effectiveness in [24], where it was used to suppress oscillations for the slewing motion of the HMF 2020K4 loader crane. The load pressure is defined as the effective pressure acting on a cylinder—see Equation (52)—and is derived from the *a*-side and *b*-side pressures, respectively. By measuring these pressures, the load pressure can be calculated. Further, by measuring the position and velocity of the cylinder, the gravitational term $G(x)$ and the friction term $F_{fric}$ can be estimated. Using proper filtering, the acceleration $\ddot{x}$ can be estimated based on Equation (53). The inertia term $M(x)$ represents the effective mass of the cylinder.

$$p_L = p_a - \frac{A_b}{A_a} \cdot p_b \tag{52}$$

$$M(x) \cdot \ddot{x} = p_L \cdot A_a - G(x) - F_{fric}(\dot{x}) \tag{53}$$

### 4.1. Crane Natural Frequency

In the laboratory, the oscillations of the crane tip are measured using the Leica Absolute Tracker AT960. The hanging load is rapidly lifted for a short distance to induce oscillations in the crane, similar to an impulse response. Figure 15 shows the crane tip $z$-position in the laboratory with two load impulses for $x_t = 3.288$ m at $t = 2$ s and $x_t = 2.906$ m at $t = 32$ s.

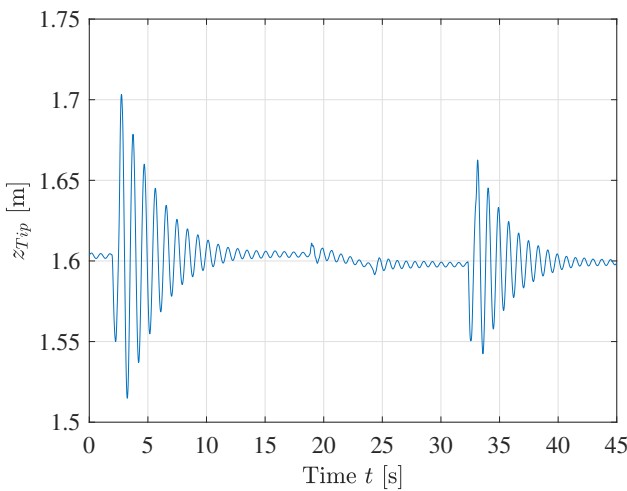

**Figure 15.** Crane tip oscillations from laboratory.

The natural frequency is extracted from the time series data, and by taking multiple measurements with varying telescopic lengths, the natural frequency of the crane tip is estimated using curve fitting. The measured and estimated crane tip natural frequency is shown in Figure 16. The formula for the estimate is given as

$$\hat{\omega}_{Tip} = 0.11 \tfrac{\text{rad/s}}{\text{m}^2} \cdot x_t^2 - 1.716 \tfrac{\text{rad/s}}{\text{m}} \cdot x_t + 11.63 \text{ rad/s} \tag{54}$$

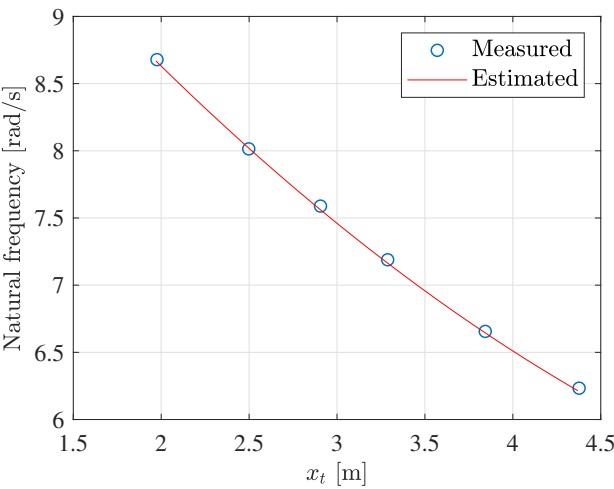

**Figure 16.** Estimated crane natural frequency.

### 4.2. Adaptive Bandpass Filter

An adaptive bandpass filter is used to extract the crane tip oscillations from the main cylinder load pressure. A critically damped bandpass filter is selected, which has the following transfer function:

$$G_{BP}(s) = \frac{2 \cdot s \cdot \omega_f}{s^2 + 2 \cdot s \cdot \omega_f + \omega_f^2} \tag{55}$$

$$= 2 \cdot \frac{\omega_f}{s + \omega_f} \cdot \frac{s}{s + \omega_f} \tag{56}$$

The bandpass filter is used in a feedback loop to suppress the oscillations. The control signal from the dynamic compensator with feedback gain $k_{pL}$ is then given as

$$u_{comp} = k_{pL} \cdot p_L \cdot G_{BP}(s) \tag{57}$$

A digital implementation with variable bandpass frequency is made by separating the bandpass filter into a lowpass filter and a highpass filter, shown in Equations (58)–(62). The estimated eigenfrequency of the crane is used as the center frequency of the filter. $y_{HP}$, $y_{LP}$, and $y_{BP}$ denote the output of the highpass, lowpass, and bandpass filters, respectively.

$$\omega_f = \hat{\omega}_{Tip}(x_t) \tag{58}$$

$$\alpha = \frac{1}{1 + \omega_f \cdot T_s} \tag{59}$$

$$y_{HP,i} = \alpha \cdot y_{HP,i-1} + \alpha \cdot (x_i - x_{i-1}) \tag{60}$$

$$y_{LP,i} = \alpha \cdot y_{LP,i-1} + (1 - \alpha) \cdot y_{HP,i} \tag{61}$$

$$y_{BP,i} = 2 \cdot y_{LP,i} \tag{62}$$

where

$i$ = sample number;
$x$ = filter input;
$y$ = filter output(s);
$T_s$ = sample time, 0.01 s;
$\hat{\omega}_{Tip}$ = estimated tip eigenfrequency.

## 5. Modeling of Telescopic Actuation System

A model of the crane containing the hydraulic system, main boom, and knuckle boom was previously created using Simscape™ components; see [23–25]. For this paper, the telescopic actuation system is modeled and added to the Simscape model. A section view from the CAD model is given in Figure 17, showing how the telescopic booms are packed inside each other, as well as showing the unique telescopic cylinders. An illustration of a telescopic cylinder is given in Figure 18, showing how the three tubes are used to transport the fluid through the telescopic system. The associated diameters $D$ and stroke length $h$ are given in Table 5. Note that $D_{i,m}$ refers to the inner diameter of the middle tube, etc.

**Table 5.** Telescopic cylinder data, in [mm].

|            | $D_{o,o}$ | $D_{i,o}$ | $D_{o,m}$ | $D_{i,m}$ | $D_{o,i}$ | $D_{i,i}$ | $h$  |
|------------|-----------|-----------|-----------|-----------|-----------|-----------|------|
| Cylinder 1 | 80        | 70        | 55        | 35        | 20        | 15        | 1885 |
| Cylinder 2 | 80        | 70        | 55        | 35        | 20        | 15        | 1950 |
| Cylinder 3 | 80        | 70        | 50        | 34        | 20        | 15        | 2000 |
| Cylinder 4 | 80        | 70        | 50        | 34        | 20        | 15        | 2100 |

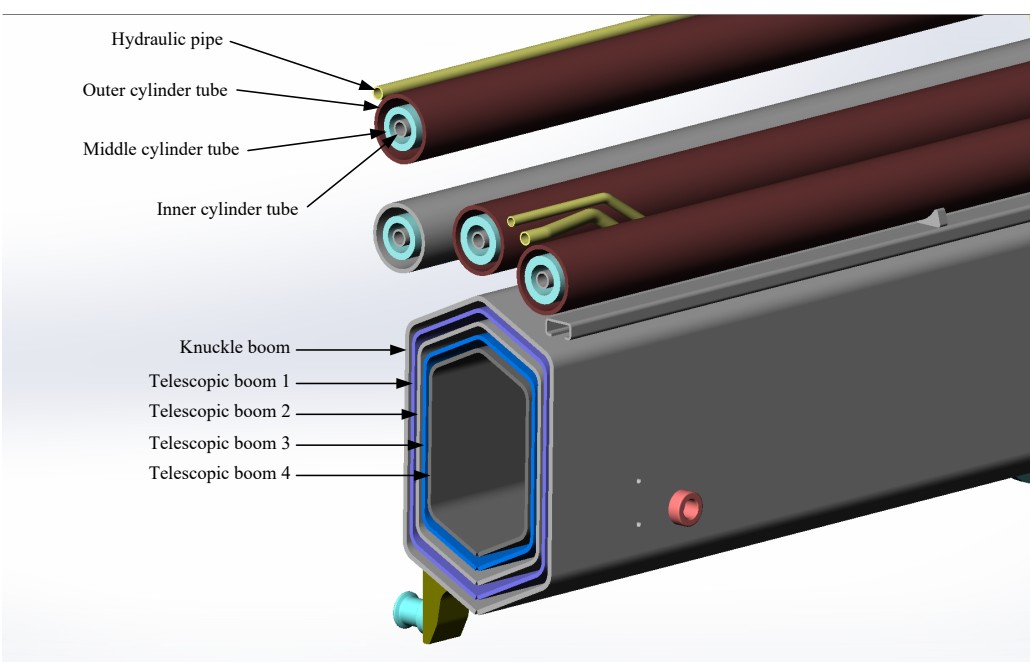

**Figure 17.** Section view of the telescopic system, from CAD model.

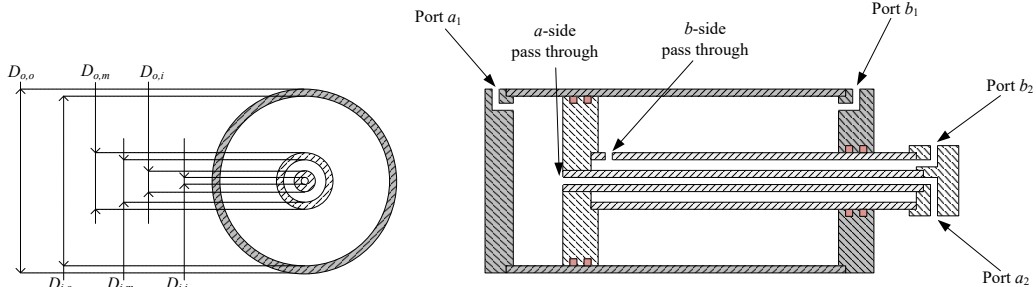

**Figure 18.** Illustration of a telescopic cylinder.

The inner tube allows for the *a*-side of all cylinders to be connected together, effectively making the cylinders connected in parallel, hydraulically. Mechanically, the booms are connected in series. The effective area and area ratio $\phi$ of the cylinders are given by

$$A_a = \frac{\pi}{4} \cdot D_{i,o}^2 \tag{63}$$

$$A_b = \frac{\pi}{4} \cdot D_{i,o}^2 - \frac{\pi}{4} \cdot D_{o,m}^2 \tag{64}$$

$$\phi = \frac{A_b}{A_a} \tag{65}$$

The hydraulic system for the telescopic cylinders is shown in Figure 19. The counterbalance valve is a special recirculating type, which effectively connects the *a*-side and *b*-side to the same pressure during the out-stroke motion. The workings of the telescopic circuit are shown in detail in Figure 20. Since the outer diameter of the middle tube is larger for cylinders 1 and 2, they have a smaller *b*-side area and will start moving first. The counterbalance valve has a rated flow of 70 L/min, and a pilot area ratio of 4. The directional control valve has a rated flow of 40 L/min.

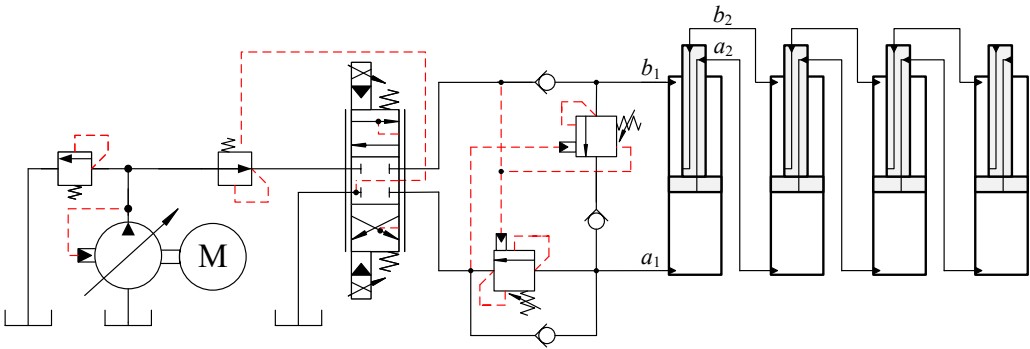

**Figure 19.** Hydraulic system for the telescopic cylinders.

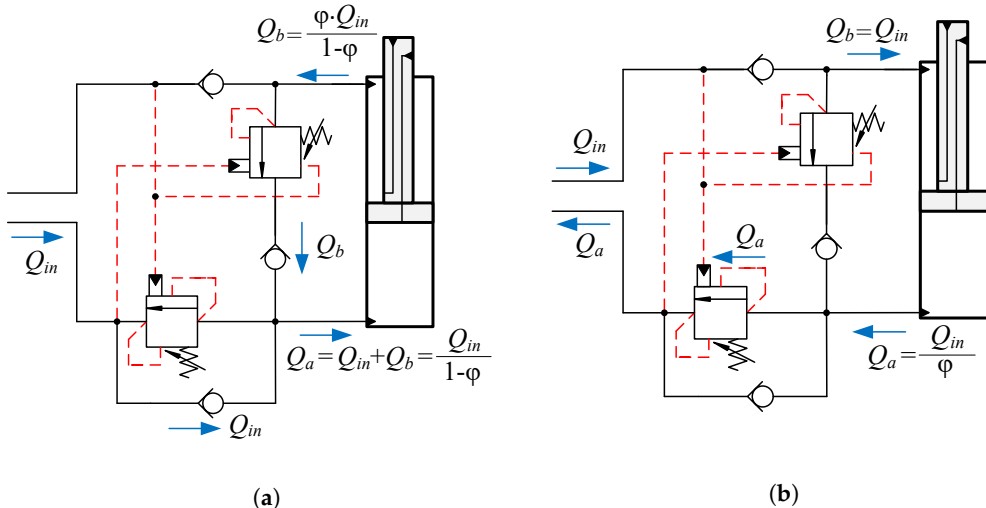

(**a**)　　　　　　　　　　　　　　　　　　　　　(**b**)

**Figure 20.** Flows in the telescopic circuit. (**a**) Flows in the telescopic circuit during out-stroke motion. (**b**) Flows in the telescopic circuit during in-stroke motion.

To obtain the proper motion sequence of the telescopic cylinders as observed on the physical crane, the friction in each is adjusted such that the outer cylinders have slightly more friction. The inner cylinder has an estimated Coulomb friction of 10 kN and viscous friction of 2 kN· s/m. The outer cylinders have an adjusted viscous friction of 2.1, 2.2, and 2.3 kN· s/m, respectively. Figure 21 shows that the inner cylinder moves first, as is desired. It can also be seen that the velocities of cylinders 3 and 4 are slightly higher since they have a larger area ratio $\phi$, which results in a higher $Q_a$ during the out-stroke motion.

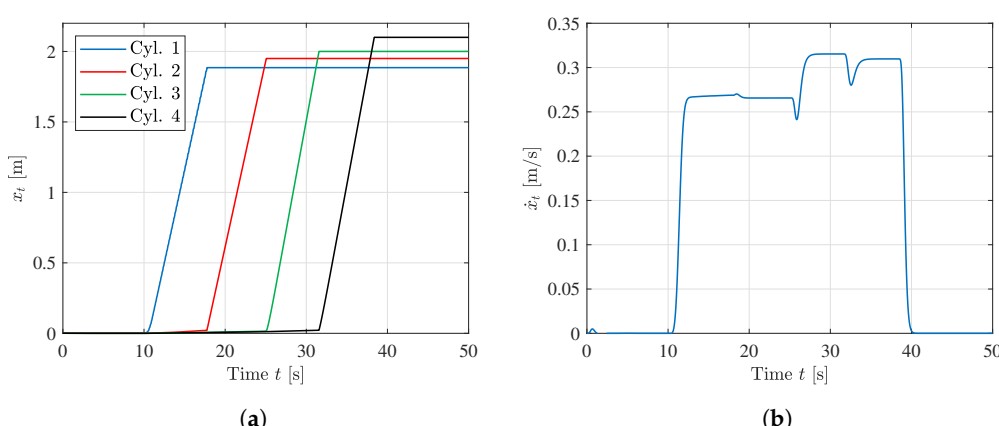

(**a**)　　　　　　　　　　　　　　　　　　　　　(**b**)

**Figure 21.** Motion sequence and velocity of the telescopic cylinders. (**a**) Position of each telescopic cylinder. (**b**) Velocity of the full telescopic system.

## 6. Simulation Results

To verify the performance and feasibility of the static and dynamic deflection compensator, simulations are performed in the MATLAB Simulink® environment. A simplification of the flexibility is made by placing a rotational spring between the knuckle boom and first telescopic boom, illustrated in Figure 22. A similar approach was used in [26]. The deflection on the physical crane is due to many factors, such as structural flexibility, deformation of the sliding blocks between the booms, slack between the booms, and compression of the liquid in the cylinders. As such, an accurate model corresponding to the measured data is difficult to create, which is why a neural network is used to estimate the deflection in the first place. The purpose of the simplified model is to give the crane model an approximate static and dynamic deflection on which the developed compensator will be tested. Note that the neural network is simply re-trained to fit the measurements from the Simscape model in this section.

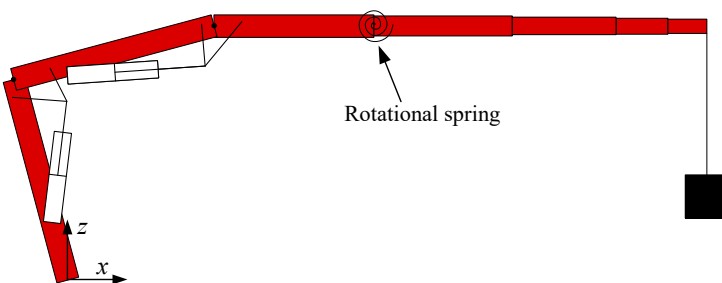

**Figure 22.** Illustration of the simplified flexible model.

In the simulations, the crane is running path control, developed earlier in [22]. Three simulations are performed, one without load, one with load, and one with load and deflection compensation. A load impulse is done at $t = 20$ s. The crane is moving from $[x_m, x_k, x_t]^T = [1.38 \text{ m}, 1.8 \text{ m}, 4 \text{ m}]^T$ to $[1.43 \text{ m}, 1.85 \text{ m}, 2 \text{ m}]^T$. Figure 23 shows the vertical position $z_{Tip}$ during simulations, as the deflection is largest in the $z$-direction. It can be seen that the static deflection is compensated for after 6 s. The dynamic deflection compensator quickly dampens the oscillations induced by the load impulse.

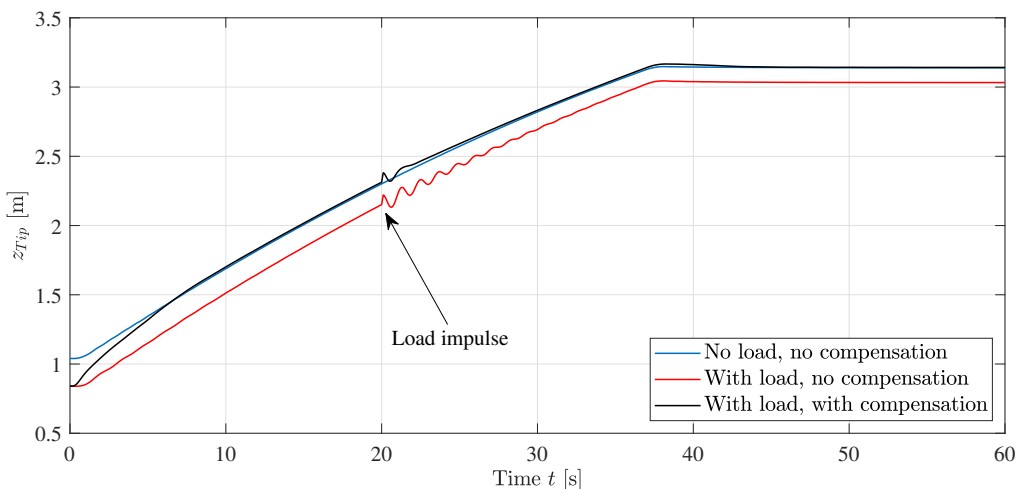

**Figure 23.** The vertical coordinate of the crane tip, $z_{Tip}$, is plotted as a function of time for three different conditions.

To investigate the effects of the deflection compensator, the change in cylinder position reference $\Delta x_{ref} = \hat{x}_{ref} - x_{ref}$ from the static compensator and the control signal $u_{comp}$ from the dynamic compensator is shown in Figure 24. The static deflection compensator modifies

the cylinder position reference by a few centimeters, varying smoothly with the cylinder positions. The effect of the dynamic deflection compensator is most prominent at the load impulse at $t = 20$ s, giving a rapid correction to dampen the oscillations.

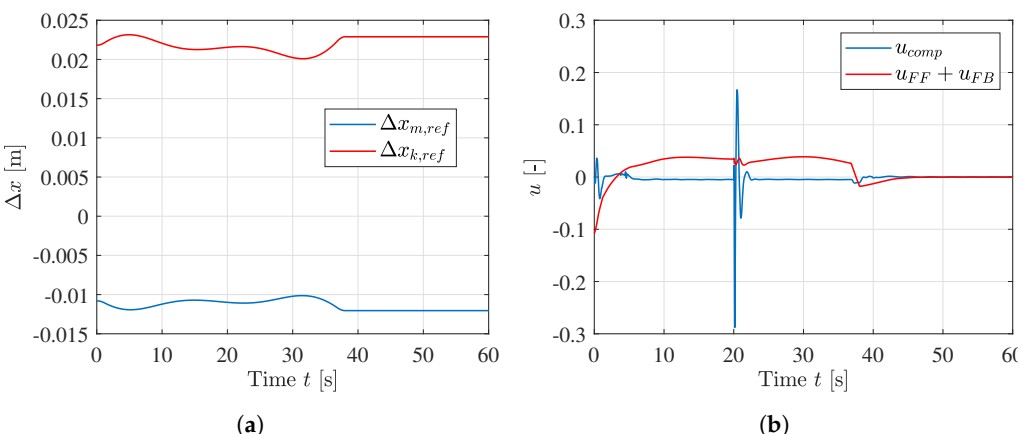

(**a**)                                                     (**b**)

**Figure 24.** Effects of the static and dynamic deflection compensator. (**a**) Change in cylinder position reference from the static deflection compensator, (**b**) control signal from the dynamic deflection compensator $u_{comp}$, and motion controller $u_{FF} + u_{FB}$.

## 7. Experimental Verification

Experiments are conducted on the HMF 2020K4 loader crane in the laboratory. The deflection compensator is implemented on a CompactRIO connected to the crane. A picture of the test setup is shown in Figure 25.

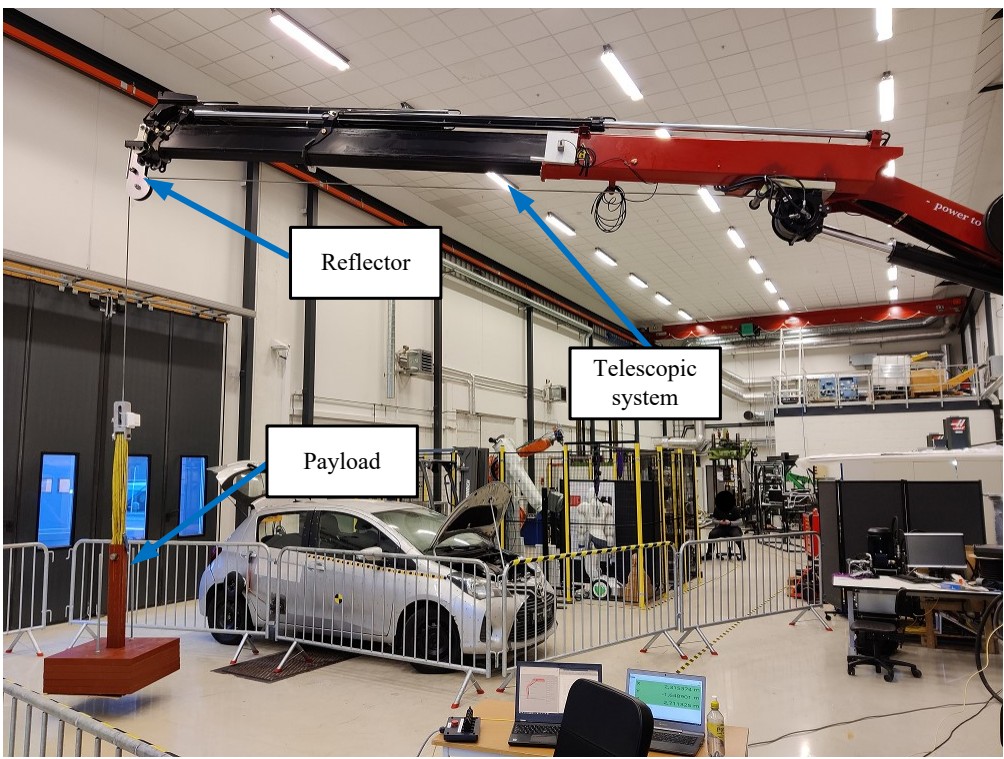

**Figure 25.** Experimental setup in the laboratory, showing the crane with a hanging load.

There is some deadband in the valves on the loader crane. Deadband compensation is implemented for the laboratory experiments. The deadbands are identified from the test and are shown in Table 6. The equation for the deadband compensator is shown

in Equation (66). A small transition region $\tilde{u}$ is introduced to keep the control signal continuous, which reduces unnecessary oscillations, and ensures that the valve will be able to stay closed when no movement is needed. Parameters used in the experiments are shown in Table 7.

**Table 6.** Identified deadband for the actuators.

| Actuator | Out, $u^+$ | In, $u^-$ |
|---|---|---|
| Main | 0.24 | −0.22 |
| Knuckle | 0.20 | −0.31 |
| Telescope | 0.33 | −0.33 |

$$
\hat{u} = \begin{cases} \min\left(u^+ + (1-u^+)u, \dfrac{u^+}{\tilde{u}}u\right) & \text{if } u > 0 \\ \max\left(u^- + (1+u^-)u, -\dfrac{u^-}{\tilde{u}}u\right) & \text{else} \end{cases} \tag{66}
$$

where

$\hat{u}$ = compensated control signal
$u$ = control signal
$u^+$ = out-stroke deadband
$u^-$ = in-stroke deadband
$\tilde{u}$ = transition region, 0.01.

**Table 7.** Parameters used in laboratory.

| Description | Name | Value |
|---|---|---|
| Main feedback | $k_{p,m}$ | 15 m$^{-1}$ |
| Main out-stroke feedforward | $k_{ff,m}^+$ | 30.16 s/m |
| Main in-stroke feedforward | $k_{ff,m}^-$ | 18.37 s/m |
| Knuckle feedback | $k_{p,k}$ | 20 m$^{-1}$ |
| Knuckle out-stroke feedforward | $k_{ff,k}^+$ | 26.51 s/m |
| Knuckle in-stroke feedforward | $k_{ff,k}^-$ | 14.72 s/m |
| Telescope feedback | $k_{p,t}$ | 2 m$^{-1}$ |
| Telescope out-stroke feedforward | $k_{ff,t}^+$ | 3.33 s/m |
| Telescope in-stroke feedforward | $k_{ff,t}^-$ | 3.7 s/m |
| Pressure feedback gain | $k_{pL}$ | 0.02 bar$^{-1}$ |

The crane moves from $[x_m, x_k, x_t]^T = [0.395 \text{ m}, 0.6151 \text{ m}, 4.168 \text{ m}]^T$ to $[0.4869 \text{ m}, 0.6161 \text{ m}, 2.015 \text{ m}]^T$ while using the path controller. The path is run three times, one without load, one with load, and one with load and deflection compensation. A plot of the tip position in the $xz$-plane is shown in Figure 26, while the $z$-position versus time is shown in Figure 27. It can be seen that the compensator is able to compensate for the static deflection almost completely, in addition to removing the oscillations at the start and end of the motion. At the end of the path, the static deflection of the vertical coordinate of the crane tip, $z_{Tip}$, is reduced from 56.8 mm to 5.7 mm, a 90% decrease.

To showcase the effects of the deflection compensator, the change in cylinder position reference $\Delta x_{ref} = \hat{x}_{ref} - x_{ref}$ from the static compensator and the control signal $u_{comp}$ from the dynamic compensator is shown in Figure 28. The static deflection compensator modifies the cylinder position reference smoothly. The effect of the dynamic deflection compensator can be seen throughout the whole motion, actively suppressing oscillations.

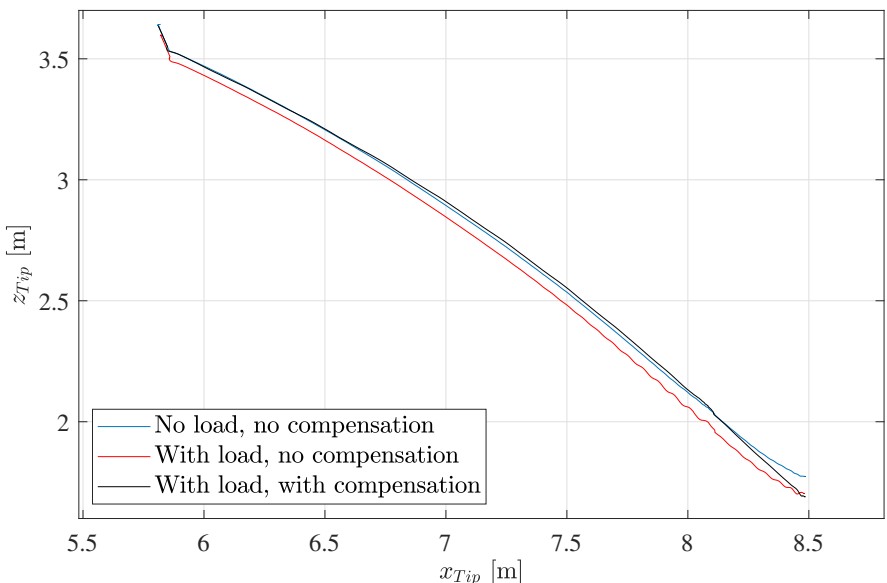

**Figure 26.** Crane tip position in the *xz*-plane during motion in laboratory.

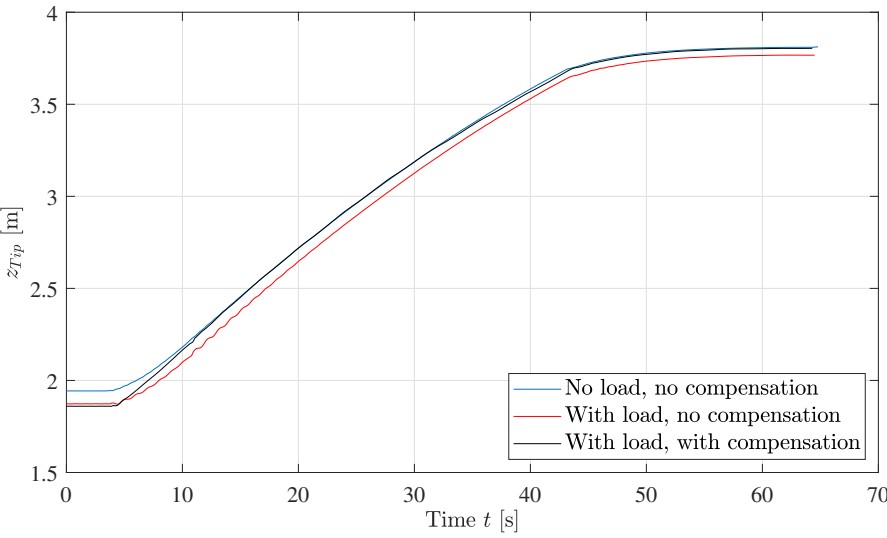

**Figure 27.** The vertical coordinate of the crane tip, $z_{Tip}$, plotted as a function of time during motion in laboratory.

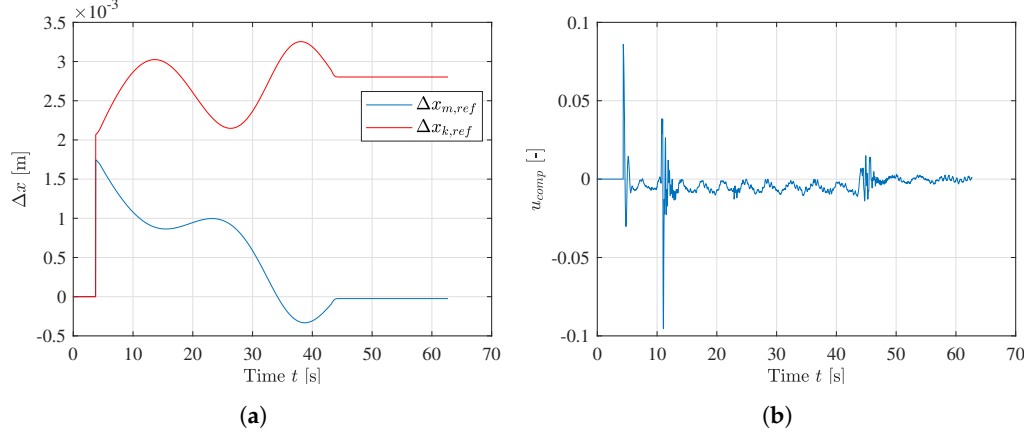

**Figure 28.** Effects of the static and dynamic deflection compensator in laboratory. (**a**) Change in cylinder position reference from the static deflection compensator. (**b**) Control signal from the dynamic deflection compensator.

To demonstrate the capabilities of the dynamic deflection compensator more clearly, a load impulse test is performed similar to the simulations. In this case, the position controller is disabled and only the dynamic compensator is activated. It can be seen in Figure 29 that the dynamic deflection compensator quickly dampens the oscillations. The slight drift at the end occurs simply because the position controller is deactivated during this test.

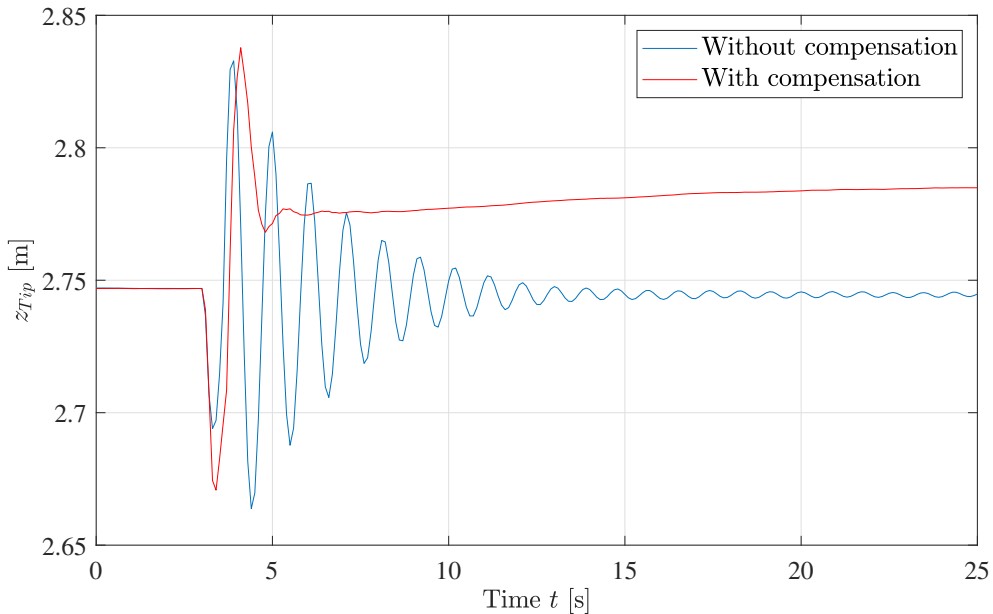

**Figure 29.** Load impulse test with only dynamic compensator activated.

## 8. Conclusions

In this paper, a novel method for stable deflection compensation is introduced and implemented on a commercially available loader crane. The new method is verified both numerically via simulations and experimentally by conducting several path-following tasks that all clearly demonstrate the simultaneous effect of compensating for crane tip deflections while suppressing oscillations in the system. The deflection of the crane tip is first measured in the laboratory using a laser tracker. A neural network deflection estimator is designed and trained using backpropagation with training data from laboratory measurements. The relevant kinematic functions are derived for the static compensator and are used to transform the cylinder position reference from the actuator space to Cartesian coordinates. The estimated tip deflection is added to the reference, and inverse kinematics are then used to transform the modified reference from Cartesian space into actuator space. The dynamic compensator uses pressure feedback with an adaptive bandpass filter to extract the crane tip oscillations while allowing for steady-state variations. This signal is then used in a feedback loop to compensate for these oscillations.

Simulation results show that the static compensator is able to minimize the effects of the deflection and move the crane tip to a similar position as in the no-load case. The dynamic compensator is able to suppress the oscillations in both general path traveling as well as load impulse situations.

Laboratory experiments are conducted to evaluate the control system on the hydraulic loader crane in practice. Experimental results are similar to the simulations in that both the static and dynamic compensators are able to minimize the effects of the deflection and oscillations. The crane tip is able to follow the same position as in the no-load case with a 90% decrease in static deflection. In the laboratory, the control signal from the dynamic compensator successfully suppresses the oscillations during the entire motion.

Further work may include stability analysis of the neural network deflection estimator and adaptive bandpass filter. Since the system was tested with a heavy payload, the ef-

fects of lighter loads may also be investigated, for example, by adding another input to the deflection estimator representing the measured or estimated weight of the payload. Different types of neural networks may also be investigated, for example, dynamic neural networks. In addition, the novel method requires a mapping of the deflection and an estimate of the natural frequency of the crane. While the best results are obtained by physical measurements as presented in this paper, it is expected that simpler and less time consuming estimates can still yield significant improvement in accuracy and stability when implemented using the presented method.

**Author Contributions:** Conceptualization, K.J.J., M.K.E. and M.R.H.; methodology, K.J.J.; software, K.J.J.; validation, K.J.J.; formal analysis, K.J.J.; investigation, K.J.J.; data curation, K.J.J.; writing—original draft preparation, K.J.J.; writing—review and editing, K.J.J., M.K.E. and M.R.H.; visualization, K.J.J.; supervision, M.K.E. and M.R.H. All authors have read and agreed to the published version of the manuscript.

**Funding:** This research was funded by the Norwegian Ministry of Education and Research grant number 155597. The APC was funded by the University of Agder.

**Conflicts of Interest:** The authors declare no conflict of interest.

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
