# Peer review of "Online Deflection Compensation of a Flexible Hydraulic Loader Crane Using Neural Networks and Pressure Feedback"

_robotics, doi:10.3390/robotics11020034_

Round 1
Reviewer 1 Report
This paper is well written and can be accepted after minor revisions. Comments are listed following: The crane considered in this work has a similar mechanism to robot arms and works on using neural networks for robot arm control may be of reference values to this paper, e.g., distributed recurrent neural networks for cooperative control of manipulators: a game-theoretic perspective,kinematic control of redundant manipulators using neural networks,A novel recurrent neural network for manipulator control with improved noise tolerance,A dynamic neural network approach for efficient control of manipulators. Please discuss the possibility to use above dynamic neural networks for the control of the crane and possible pros and cons in comparison with the presented approach in the introduction part. Please provide detailed data for the experimental results. Also, preferably, a demonstration video will help reviewers to see the real performance.
Author Response
See PDF file

Reviewer 2 Report
This manuscript proposes a novel method for deflection compensations using based on a shallow neural network.
Minor comments, suggestions:
- Abstract and conclusion should contain quantitative results: show how the proposed method performs to other possible methods;
- The proposed neural network for deflection estimation is a classical architecture, the thorough description is unnecessary;
- Have you considered other machine learning based methods for estimation? Did you compare to the presented MLP?
- Did you experiment with hyperparameter tuning, e.g. number of hidden layer neurons, activation, different regularization methods and parameters?
In my opinion, it is really good to see that the simulated results were verified in a real life experiment, the manuscript is a lot more confident.
Author Response
See PDF file

Reviewer 3 Report
1. This well-written paper presents an interesting deflection compensation scheme for a hydraulic loader crane. The authors offer thorough kinematics and actuator analysis and show the neural network deflection estimator.
2. It will be nice to see more experimental results conducted on the hydraulic loader crane.
Author Response
See PDF file
